# Impact of COVID-19 on outpatient appointments in children and young people in England: an observational study

Alex Bottle ⬤ ,[1] Francesca K Neale ⬤ ,[1] Kimberley A Foley ⬤ ,[1] Russell M Viner ⬤ ,[2] Simon Kenny ⬤ ,[3,4] Paul Aylin ⬤ ,[1] Sonia Saxena ⬤ ,[1] Dougal S Hargreaves ⬤ [1]

¹Department of Primary Care and Public Health, Imperial College London, London, UK
²Population, Policy and Practice Research Programme, UCL Great Ormond Street Institute of Child Health Population Policy and Practice, London, UK
³National Clinical Director, NHS England and NHS Improvement, London, UK
⁴Institute of Systems, Molecular and Integrative Biology, University of Liverpool, Liverpool, UK

**Correspondence to**
Dr Francesca K Neale;
francesca.neale1@nhs.net

## ABSTRACT

**Objectives** To describe the impact of the COVID-19 pandemic on outpatient appointments for children and young people.

**Setting** All National Health Service (public) hospitals in England.

**Participants** All people in England aged <25 years.

**Outcome measures** Outpatient department attendance numbers, rates and modes (face to face vs telephone) by age group, sex and socioeconomic deprivation.

**Results** Compared with the average for January 2017 to December 2019, there was a 3.8 million appointment shortfall (23.5%) for the under-25 population in England between March 2020 and February 2021, despite a total rise in phone appointments of 2.6 million during that time. This was true for each age group, sex and deprivation fifth, but there were smaller decreases in face to face and total appointments for babies under 1 year. For all ages combined, around one in six first and one in four follow-up appointments were by phone in the most recent period. The proportion of appointments attended was high, at over 95% for telephone and over 90% for face-to-face appointments for all ages.

**Conclusions** COVID-19 led to a dramatic fall in total outpatient appointments and a large rise in the proportion of those appointments conducted by telephone. The impact that this has had on patient outcomes is still unknown. The differential impact of COVID-19 on outpatient activity in different sociodemographic groups may also inform design of paediatric outpatient services in the post-COVID period.

## STRENGTHS AND LIMITATIONS OF THIS STUDY

⇒ This paper benefits from national data collected in a standardised way.
⇒ The data quality for administrative items like dates, age and sex is very high.
⇒ We were unable to analyse patterns by diagnosis due to very low levels of recording.
⇒ Although we examined several patient subgroups, future work could look at patterns by hospital trust and specialty.

attendance and emergency admissions have been documented in both children[1–8] and adults.[9] A national retrospective cohort study in Scotland described reductions of almost two-thirds in paediatric unscheduled primary care presentations and half of emergency secondary care presentations, without any change in paediatric intensive care unit (PICU) admissions or mortality.[6] Falls in urgent care contacts compared with previous years led to speculation that CYP health might deteriorate if serious conditions were missed or if they experienced delays in diagnosis of chronic disease. However, studies have found no significant impact on route to diagnosis and severity of presentation for type 1 diabetes and new diagnosed cancers, as well as no impact on overall outcomes for asthma, pyloric stenosis and appendicitis.[10–14]

There have also been concerns that some CYP were unable to access specialist services in hospitals. For example, a study of radiology outpatient activity at a tertiary children's hospital from March 2020 to March 2021 showed a reduction of 27.3%.[15] The British Medical Association estimated that across all ages in England between April and June 2020 there were between 2.47 million and 2.6 million fewer first outpatient attendances than expected.[16] Similarly, The Health Foundation reported a fall of 4.4 million outpatient

## INTRODUCTION

The COVID-19 pandemic has led to profound disruption to health services globally as treatment of COVID-19 infections was prioritised over some groups felt to be at lower risk, including children and young people (CYP). In the UK, strict lockdown measures were introduced in March 2020, November 2020 and January 2021. The disruptions to daily life and health services, including the temporary closure of some paediatric wards and emergency departments, led to changes to health seeking behaviour and health service use. Reductions in emergency department

**Table 1** Changes in numbers of scheduled appointments by age group for March 2020 to February 2021 compared with yearly average across the period January 2017 to December 2019

| Age (years) | First F2F appt | Follow-up F2F appt | First phone appt | Follow-up phone appt | Total (% change) |
|---|---|---|---|---|---|
| Under 1 | −167 106 (−28.1) | −124 745 (−27.1) | +50 189 (+542.4) | +69 267 (+311.0) | −172 395 (−15.9%) |
| 1–4 | −408 050 (−46.6) | −635 432 (−42.6) | +104 189 (+881.6) | +318 574 (+675.2) | −620 819 (−25.6%) |
| 5–9 | −424 378 (−47.5) | −819 892 (−43.1) | +103 778 (+514.9) | +395 336 (+639.2) | −745 056 (−25.9%) |
| 10–14 | −403 763 (-46.8) | −786 708 (−41.8) | +97 933 (+317.5) | +384 810 (+585.7) | −707 728 (−24.9%) |
| 15–19 | −388 535 (−42.4) | −835 778 (−39.5) | +116 457 (+412.0) | +383 934 (+559.8) | −723 922 (−23.1%) |
| 20–24 | −497 072 (–37.4) | −879 924 (−36.2) | +159 949 (+523.9) | +378 020 (+540.5) | −839 027 (−21.7%) |
| Total | −2 288 804 (–41.8) | −4 082 479 (–39.7) | +632 395 (+483.3) | +1 929 941 (+575.2) | −3 808 947 (−23.5%) |

'First' and 'follow-up' are defined by HES.
F2F, face-to-face; HES, Hospital Episodes Statistics.

appointments in England in May 2020 compared with May 2019 and 4 million fewer general practitioner referrals to outpatient appointments between January and October 2020 compared with the same period in 2019, although referrals for suspected cancer had recovered to prepandemic levels by October 2020.[17] However, no detailed analysis has been published of paediatric outpatient activity, where growing concerns had already been raised about the equity of access to specialist care pre-COVID.[18]

Telephone and video consultations have been used prior to the COVID-19 pandemic in many specialties, including paediatrics, with advantages including less time wasted between appointments, no associated costs, fewer late arrivals and, since the COVID-19 pandemic, patients are understandably anxious about the risk of catching COVID-19 in hospital. Disadvantages of remote consultations include missing non-verbal cues, the exclusion of those without access to technology, the inability to examine the patient, and that many doctors have not been trained to conduct telephone consultations.[19–21] However, it is unknown to what extent remote paediatric consultations were used during the COVID-19 pandemic.

We make use of England's national hospital administrative database, Hospital Episodes Statistics (HES), to describe trends in outpatient appointment numbers, rates and modes (face to face vs telephone) in CYP by age group, sex and socioeconomic deprivation between January 2017 and January 2021 in England.

## METHODS
### Data
We extracted HES data for outpatient appointments between 1 January 2017 and 28 February 2021 and for patients aged under 25. Records were excluded if either the patient or the hospital cancelled the appointment (between 7.7% and 10.4% of appointments in 2019–2020, eg, depending on the age group) or in the rare cases when the age or sex was invalid. Area-level Index of Multiple Deprivation (IMD) was attached to HES for each patient and turned into population-weighted fifths so that nationally, for all ages combined, each fifth

contained equal population. The ATENTYPE field distinguishes between first and follow-up appointments, with separate values for face to face, telephone and not specified; we excluded records for 'not specified' (0.2%–0.6% of appointments for the whole period, depending on the age group) when comparing face to face and telephone. It also states whether the patient attended. The term telephone consultations refers to both telephone and video consultations as no separate value for video consultations is available.

The prepandemic period was defined as 1 January 2017 to 29 February 2020; the pandemic period was defined as 1 March 2020 onwards.

### Statistical analysis
Appointment counts and attendance rates (ie, proportion of appointments that the patient attended) were plotted by month and patient subgroup, including by age (under-1, 1–4, 5–9, 10–14, 15–19 and 20–24). For ease of comparison, changes in appointment numbers were calculated for the 12-month period from 1 March 2020 to 28 February 2021 rather than to the end of Mar 2021. This was done by comparing total counts with age-matched yearly averages across the 3-year period 1 January 2017 to 31 December 2019 rather than use time-series predictions, for example, as no clear monthly trends were noted during that time. Changes were expressed in absolute terms and as percentages. Due to the descriptive rather than hypothesis-driven nature of our study and the very large numbers involved, no statistical tests were performed.

### Patient and public involvement
There was no patient or public involvement in this research.

## RESULTS
For CYP aged under 25 years, the overall fall in outpatient appointments for the 12-month period, March 2020 to February 2021 compared with the means for the previous 3 years was over 3.8 million, a 23.5% fall, with the biggest

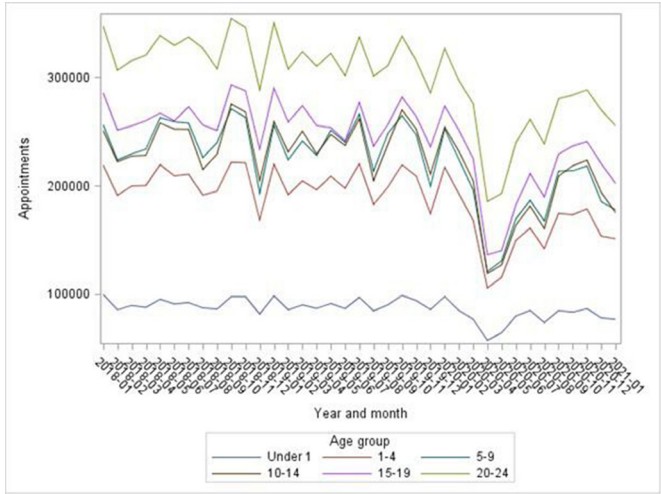

**Figure 1** All scheduled appointments since January 2018 by age group.

percentage fall in the 5–9s and the smallest percentage change in under 1s (table 1). This was despite a total rise in phone appointments of 2.6 million.

Total outpatient appointment numbers increased from May 2020 but had not reached prepandemic levels by Mar 2021 (figure 1). This was true for each age group and sex (online supplemental figure S1) and for surgical and non-surgical specialties (online supplemental figure S2). The same pattern was observed for each IMD fifth (online supplemental figure S3, S4), which was unchanged since March 2020.[10] To illustrate, figure 2 shows this for the under-1s and those aged 20–24: for all age groups, the less-affluent areas had the most appointments.

Phone appointments increased by 15% from Mar 2020 to February 2021 for both first and follow-up visits. Table 2 compares prepandemic (January 2017 to December 2019) averages with October 2020 to February 2021 averages. The largest relative changes were seen among under 1 s with tripling of the proportion of first appointments

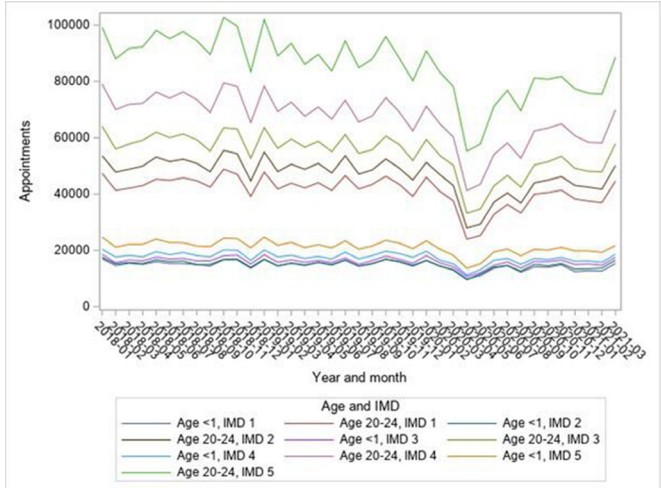

**Figure 2** Trends in total appointments by IMD fifth for ages 0–1 and 20–24; IMD 1 is least deprived, IMD 5 is most deprived. IMD, Index of Multiple Deprivation.

performed by phone (1.5%–4.6%) and almost doubling of the proportion of follow-up appointments performed by phone (10.5%–20.0%). The largest percentage rises were for the 10–14s (table 2).

For all ages combined, around one in six first and one in four follow-up appointments were by phone in the most recent period. Online supplemental figure S5 shows the rise in phone appointments by age relative to the prepandemic period.

### Proportions of appointments attended

Attendance was high, at over 95% for telephone and over 90% for face-to-face appointments for all ages. There were slight falls in the former and slight rises in the latter from March 2021 (online supplemental figure S6, S7). Attendance and patterns did not differ by sex, with the exception that in the 20–24s, attendance was higher for women than for men, a difference driven by face-to-face appointments.

## DISCUSSION
### Summary of results

In England's National Health Service (NHS), CYP outpatient appointments fell sharply in March 2020 and had still not recovered to pre-pandemic levels in March 2021. In total, the 12 months from March 2020 saw 3.8 million appointments fewer (23.5%) than expected from the average for 2017–2019. The fall affected all ages, sexes, specialties (categorised as surgical or non-surgical), and deprivation quintiles. The huge rise in telephone use did not compensate for the bigger fall in face-to-face visits. Around one in five appointments were by phone for this population at the end of the study period.

Attendance rates were high and were higher for phone appointments than for face-to-face ones; the only sex difference was that women aged 20–24 had higher rates than men; for first phone appointments, this sex difference disappeared and was much reduced for follow-ups. In contrast to research showing persistent and systemic child health inequalities in the UK,[22] there was little variation in appointment numbers by IMD quintile.

### Findings compared with previous studies

Our findings fit with other national research that demonstrated large reductions in secondary care health service use since March 2020,[6 9] including reduced outpatient activity for all age groups[17] Whereas total suspected cancer referrals returned to prepandemic levels by October 2020, we found that CYP outpatient activity had not returned to prepandemic levels at any point by Mar 2021.

Since the COVID-19 pandemic, there has been a significant rise in virtual consultations in CYP outpatient departments globally. A large global survey of healthcare professionals (HCPs) caring for CYP with diabetes found that 50% had switched to virtual consultations,[23] 36 of 38 surveyed North American Paediatric surgical units had switched to virtual outpatient appointments,[24]

**Table 2** Percentage of first and follow-up appointments that are by telephone, stratified by age and period (raw numbers are included in online supplemental table S1)

| Age (years) | January 2017 to December 2019 average | | October 2020 to February 2021 average* | |
|---|---|---|---|---|
| | First appt % by phone | Follow-up appt % by phone | First appt % by phone | Follow-up appt % by phone |
| Under-1 | 1.5 | 10.5 | 4.6 | 20.0 |
| 1–4 | 1.3 | 16.5 | 3.1 | 26.3 |
| 5–9 | 2.2 | 17.2 | 3.1 | 25.8 |
| 10–14 | 3.5 | 18.8 | 3.4 | 26.0 |
| 15–19 | 3.0 | 19.8 | 3.1 | 23.8 |
| 20–24 | 2.2 | 18.5 | 2.8 | 22.6 |
| Total | 2.3 | 17.3 | 3.2 | 24.5 |

*October 2020 to February 2021 is given because the first wave of COVID-19 had ended and the system had stabilised.

a global survey of paediatric neurologists reported a large increase in outpatient virtual consultations[25] and a multicentre longitudinal observational study of Paediatric orthopaedic trauma in London also described greater use of outpatient virtual consultations.[26]

Previous studies have also found that in CYP outpatient clinics, telephone appointment attendance is higher than face-to-face appointment attendance.[27 28] Studies have shown high levels of patient satisfaction with telephone appointments during the COVID-19 pandemic[29–31] and high levels of satisfaction among HCPs providing a remote CYP neurology outpatient service.[32] However, concerns have been raised about the quality of care provided in remote consultations, especially for first appointments.[33 34] Research is emerging on how remote CYP outpatient appointments should best be routinely integrated into post-COVID services.[35–37]

The fall in CYP outpatient activity reflects changes in need, a decrease in health seeking behaviour and changes in health system pathways, including a move towards remote consultations and redeployment of Paediatric staff to care for COVID-19 patients. Similar changes in activity during this period were seen across the health system, including in primary care[38] and emergency care.[4 39]

The extent to which the fall in CYP outpatient activity can be attributed to a decrease in health seeking behaviour is unknown. A retrospective national cohort study into the decrease of paediatric secondary care emergency presentations hypothesised that, as there were no increases in clinical severity scores, PICU admissions or mortality, caregivers were often appropriately able to use a higher threshold for seeking medical attention.[6] Another study on this topic also found that 93.5% of CYP presenting to secondary care during a 2-week period in April–May 2020 were not felt to have had a delay in their presentation.[4] Further research is required to establish the extent to which the fall in CYP outpatient activity since March 2020 is a result of changes in health seeking behaviour, although it is unlikely to have played as large a role as in emergency presentations as we observed a dramatic fall in both first and follow-up appointments.

## Strengths and limitations

This is the largest study of outpatient appointment patterns in CYP before and after March 2020 in England. It benefits from national data collected in a standardised way. Data quality in such databases for administrative items like dates, age and sex is very high, but we were unable to analyse patterns by diagnosis due to very low levels of recording. Rather than precomparisons and postcomparisons, a more sophisticated analysis could be run using interrupted or other time series models, but we inspected prepandemic patterns and discerned no clear trends that needed adjustment.

We examined several patient subgroups, but future work could look at patterns by hospital trust and specialty. We saw falls in first and follow-up visits, and one could track whether clinic discharge rates changed during the period. In particular, it will be important to follow patients up to observe their future outcomes, although linkage with primary care records would be needed to capture patients who were not referred to outpatient departments.

## Implications for policy and practice

It is reassuring that despite huge changes in clinical practice, face-to-face appointments for babies under 1 year were relatively preserved, compared with the decrease in face-to-face appointments for other age groups, and that activity changes were similar among CYP living in more and less deprived areas. This suggests that the NHS continued to provide a responsive, equitable and service for CYP needing specialist care throughout the pandemic, although at a reduced level.

While part of the reduction in activity may be accounted for by reduced prevalence of infective conditions and greater parental empowerment and confidence in managing self-limiting illness,[4 40] there is likely to be significant ongoing unmet need for specialist paediatric care. Whereas the impact of delay in accessing acute care can be assessed fairly quickly using routinely collected data, the impact of reduced access to outpatient services may be reflected more in worse quality of life for CYP and

potentially worse longer-term outcomes through lack of early diagnosis and early intervention.[41] Identifying and addressing these unmet needs should be a priority for cilnicians, commissioners and policy-makers.

These findings may also usefully inform the current debate about integrated care for CYP and the best ways of combining high quality care with convenience and positive experience for patients and families. In particular, greater use of remote care and patient family-initiated follow-up have huge potential benefits as well as potential risks in lower quality care or increasing inequalities. Further research is needed to understand and quantify these risks to guide mitigation strategies.

## CONCLUSIONS

The impact on patients and the NHS of such a huge fall from prepandemic levels in outpatient appointments, including those in surgical specialties, is yet to be felt. The extent to which the fall in total appointments and the rise in the use of telephones matters in terms of patient health outcomes will need further monitoring. The natural experiment of radical changes to paediatric outpatient care during the pandemic period may also inform development of more integrated and responsive specialist services for CYP in the future.

**Contributors** AB and DSH devised the study. AB devised the approach to and conducted the data analysis. FKN and AB drafted the manuscript. All authors (AB, FKN, KAF, RMV, SK, PA, SS and DSH) contributed to the interpretation of the results and critical revision of the manuscript and approved the final draft. AB acts as a guarantor for the final manuscript.

**Funding** This study was funded by the National Institute for Health Research grant number NIHR202322 'Understanding the disruption of children and young people's health and healthcare use during and after COVID-19 to inform healthcare and policy responses.' The Dr Foster Unit is an academic unit in the Department of Primary Care and Public Health, within the School of Public Health, Imperial College London. The unit received research funding from Dr Foster Intelligence, an independent health service research organisation (a wholly owned subsidiary of Telstra), until September 2021. The Dr Foster Unit at Imperial is affiliated with the National Institute of Health Research (NIHR) Imperial Patient Safety Translational Research Centre. The NIHR Imperial Patient Safety Translational Centre is a partnership between the Imperial College Healthcare NHS Trust and Imperial College London. The Department of Primary Care & Public Health at Imperial College London is grateful for support from the NW London NIHR Applied Research Collaboration and the Imperial NIHR Biomedical Research Centre.

**Disclaimer** The views expressed in this publication are those of the authors and not necessarily those of the NIHR or the Department of Health and Social Care.

**Competing interests** None declared.

**Patient and public involvement** Patients and/or the public were not involved in the design, or conduct, or reporting, or dissemination plans of this research.

**Patient consent for publication** Not applicable.

**Ethics approval** We had approval from the Secretary of State and the Health Research Authority under Regulation 5 of the Health Service (Control of Patient Information) Regulations 2002 to hold confidential data and analyse them for research purposes (CAG ref 15/CAG/0005). We have approval to use them for research and measuring quality of delivery of healthcare, from the London - South East Ethics Committee (REC ref 20/LO/0611).

**Provenance and peer review** Not commissioned; externally peer reviewed.

**Data availability statement** Data may be obtained from a third party and are not publicly available. The pseudonymised patient data that were used for this study can be accessed by contacting NHS Digital (see https://digital.nhs.uk/services/data-access-request-service-darshttps://digital.nhs.uk/services/data-access-request-service-dars). Access to these data is subject to a data sharing agreement (DSA) containing detailed terms and conditions of use following protocol approval from NHS Digital.

**ORCID iDs**
Alex Bottle http://orcid.org/0000-0001-9978-2011
Francesca K Neale http://orcid.org/0000-0002-3316-164X
Kimberley A Foley http://orcid.org/0000-0003-3664-8100
Russell M Viner http://orcid.org/0000-0003-3047-2247
Simon Kenny http://orcid.org/0000-0002-5917-8554
Paul Aylin http://orcid.org/0000-0003-4589-1743
Sonia Saxena http://orcid.org/0000-0003-3787-2083
Dougal S Hargreaves http://orcid.org/0000-0003-0722-9847

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
