## [Reviewer comments · BMJ Open]

ARTICLE DETAILS

TITLE (PROVISIONAL)	Impact of Covid-19 on outpatient appointments in children and young people in England: an observational study
AUTHORS	Bottle, Alex; Neale, Francesca; Foley, Kimberley; Viner, Russell; Kenny, Simon; Aylin, Paul; Saxena, Sonia; Hargreaves, Dougal

VERSION 1 – REVIEW

REVIEWER	Michael Absound Guy's and St Thomas' Hospitals NHS Trust, Department of Primary Care and Public Health
REVIEW RETURNED	30-Jan-2022

GENERAL COMMENTS	This is a well written paper on an analysis of a national (England) hospital administrative database & Hospital Episodes Statistics to describe trends in outpatient appointment (numbers, rates and modes). It provides a useful addition to the literature. If available, it will be useful to describe attendances in community vs Hospital clinics and discuss differences.
---

REVIEWER	Amy Sweeny Gold Coast Hospital and Health Service, Emergency
REVIEW RETURNED	08-Feb-2022

GENERAL COMMENTS	Thank you for giving me the opportunity to review this manuscript. The effect of COVID-19 and its mitigation strategies on health care activity is an important topic to study. The COVID-19 pandemic has inadvertently showed us how we can better deliver a substantial proportion of our health care. This may be particularly advantageous for certain age groups and certain reasons for seeking care, due to the convenience of telehealth rather than in-person appointments. Strengths: Population study using NHS dataset. Reasonable approach not to do any statistical testing due to the size of the dataset Concerns: Hospital cancellations were excluded – this may be an issue, and we need to see some data on what proportion of all appointments this represented. Appt type face to face, telephone and not specified. How many appointments were not specified? Did this vary by age group? This information needs to be provided. They stated that they did not do a time series analysis due to lack of a trend. Instead they used a three-year average of historic data for 2017-2019 This is somewhat unusual, and some evidence of a lack of a trend over the 3 years would be appropriate to include (e.g. in an appendix). Minor points:
--

	Please clarify if appointment data in Table 1 and Figure 1 is scheduled appointments only, not appointments attended. Please include this information in the title. I am unfamiliar with the health system in the UK. Is a decrease in appointments due to the inability to staff them on behalf of the health system, or a decrease in demand on behalf of the public? The data suggest that appointments (I assume scheduled appointments) dropped considerably, or by nearly one-quarter. The implications of this are that the health system reduced care offered, is this correct? Some more information here would be helpful for the international audience. On page 7 line 14, please explain what you mean by absolute percentage. Reconciliation of Table 1 with Table 2 is difficult. It seems that phone appointments might be a larger proportion of all appointments based on the numbers in Table 1. It would be helpful to see the actual numbers rather than just the amount decreased or increased. The proportion of first phone appointments for under 1s remains quite low (2.8 – 4.6% of all appointments). The change from pre-pandemic appears to be overstated. But the authors then state that face to face appointments for the under 1s have been “relatively preserved” (line 41, p9), although the tables suggest that they had dropped by 28% (Table 1), and decreased (Table 2). Some clarification of their conclusions, and/or enhancing the tables to include raw numbers not just number decreased (previously suggested) would be helpful.
--	--

VERSION 1 – AUTHOR RESPONSE

Reviewer 1: If available, it will be useful to describe attendances in community vs Hospital clinics and discuss differences.	Thank you for this suggestion and we agree that this is an important issue. Unfortunately, this data is not available in the dataset. In our strengths and limitations section we mention that future work would benefit from linkage with primary care records to look at community attendances.
Reviewer 2: Hospital cancellations were excluded – this may be an issue, and we need to see some data on what proportion of all appointments this represented.	Thank you for this important point. We have now analysed the proportion of cancellations and included this in the text. The percentage of cancellations ranged from 7.7% in the under-1-year-olds to 10.4% in the 1–4-year-olds in 2019-2020. However, are unable to analyse what proportion of these were rebooked, although in our experience it is high.
Reviewer 2: Appt type face to face, telephone and not specified. How many appointments were not specified? Did this vary by age group? This information needs to be provided.	For the whole period, the percentage of not specified appointments ranged from 0.2% in the under-1-year-olds to 0.6% in the 15–19-year-olds. We have now included this information in the text.
Reviewer 2: They stated that they did not do a time series analysis due to lack of a trend. Instead they used a three-year average of historic data for 2017-2019 This is somewhat unusual, and some evidence of a lack of a trend over the 3 years would be appropriate to include (e.g. in an appendix).	Figure S1 shows trends for each age since 2018. We have performed some reanalysis and the log RR of the monthly linear trend is just -0.00006 for all ages combined (RR of 0.99994) and varied little by age. We have added this information to Figure S1.
Reviewer 2: Please clarify if appointment data in Table 1 and Figure 1 is scheduled appointments only, not appointments attended. Please include this information in the title.	It is scheduled appointments. We have amended the Table 1 and Figure S1 titles to reflect this.
Reviewer 2: I am unfamiliar with the health system in the UK. Is a decrease in appointments due to the inability to staff them on behalf of the health system, or a decrease in demand on behalf of the public? The data suggest that appointments (I assume scheduled appointments) dropped considerably, or by nearly one-quarter. The implications of this are that the health system reduced care offered, is this correct? Some more information here would be helpful for the international audience.	Thank you for this important suggestion. We have added some more information about the different reasons for the decrease in activity into the ‘Findings compared with previous studies’ section of the paper and included two new references that demonstrate similar changes in primary and emergency care.

Reviewer 2: On page 7 line 14, please explain what you mean by absolute percentage.	By absolute percentage, we mean the percentage difference between the two absolute values. We have amended the text to read 'percentage' to make it clearer for the reader.
Reviewer 2: Reconciliation of Table 1 with Table 2 is difficult. It seems that phone appointments might be a larger proportion of all appointments based on the numbers in Table 1. It would be helpful to see the actual numbers rather than just the amount decreased or increased.	Thank you for this suggestion. We have now added Table S5 into the appendix which includes the actual numbers.
Reviewer 2: The proportion of first phone appointments for under 1s remains quite low (2.8 – 4.6% of all appointments). The change from pre-pandemic appears to be overstated. But the authors then state that face to face appointments for the under 1s have been “relatively preserved” (line 41, p9), although the tables suggest that they had dropped by 28% (Table 1), and decreased (Table 2). Some clarification of their conclusions, and/or enhancing the tables to include raw numbers not just number decreased (previously suggested) would be helpful.	We have now included Table S5 in the appendix which demonstrates the raw numbers. By “relatively preserved” we mean that the fall of 28.1% was smaller than the fall for other age groups (37.4-47.5%). We have clarified this in the text.